# Continuous Fermentation by *Lactobacillus bulgaricus* T15 Cells Immobilized in Cross-Linked F127 Hydrogels to Produce D-Lactic Acid

Yongxin Guo [1], Gang Wang [1,2,*], Huan Chen [1], Sitong Zhang [1], Yanli Li [1], Mingzhu Guo [3], Juan Liu [4] and Guang Chen [1,2]

1   College of Life Science, Jilin Agricultural University, Changchun 130118, China; guo15981389011@163.com (Y.G.); chenhuan@jlau.edu.cn (H.C.); stzhang@jlau.edu.cn (S.Z.); ylli@jlau.edu.cn (Y.L.); chg61@jlau.edu.cn (G.C.)
2   Key Laboratory of Straw Biology and Utilization, Education Ministry of China, Jilin Agricultural University, Changchun 130118, China
3   Changchun Polytechnic, Changchun 130033, China; polly1124@163.com
4   Sericultural Research Institute of Jilin Province, Changchun 132013, China; liujuan807@126.com
*   Correspondence: wanggang@jlau.edu.cn; Tel.: +86-138-4487-4718

**Abstract:** Lignocellulose biorefinery via continuous cell-recycle fermentation has long been recognized as a promising alternative technique for producing chemicals. D-lactic acid (D-LA) production by fermentation of corn stover by *Lactobacillus bulgaricus* was proven to be feasible by a previous study. However, the phenolic compounds and the high glucose content in this substrate may inhibit cell growth. The immobilization of cells in polymer hydrogels can protect them from toxic compounds in the medium and improve fermentation efficiency. Here, we studied the production of D-LA by *L. bulgaricus* cells immobilized in cross-linkable F127 bis-polyurethane methacrylate (F127-BUM/T15). The Hencky stress and Hencky strain of F127-BUM/T15 was 159.11 KPa and 0.646 respectively. When immobilized and free-living cells were cultured in media containing 5-hydroxymethylfurfural, vanillin, or high glucose concentrations, the immobilized cells were more tolerant, produced higher D-LA yields, and had higher sugar-to-acid conversion ratios. After 100 days of fermentation, the total D-LA production via immobilized cells was 1982.97 ± 1.81 g with a yield of 2.68 ± 0.48 g/L h, which was higher than that of free cells (0.625 ± 0.28 g/L h). This study demonstrated that F127-BUM/T15 has excellent potential for application in the biorefinery industry.

**Keywords:** cell-recycle fermentation; F127-BUM; corn stover; biorefinery

## 1. Introduction

Lactic acid (LA), including D-lactic acid (D-LA) and L-lactic acid (L-LA), has versatile applications in the medical, cosmetics, and food industries [1]. It is also an important platform chemical for producing value-added products [2,3]. The methods used to produce LA include chemical synthesis and microbial fermentation [4]. Microbial fermentation has gradually become the main industrial production method because of its low cost and high yield of LA with high optical purity [5].

Previous studies on LA production have tried to develop fermentation methods using non-food raw materials as the substrate and to improve the fermentation efficiency [6]. In this context, lignocellulose has emerged as one of the most important raw materials for LA production. Corn stover is one of the major agricultural residues which is about 28 million tons per year in Jilin province. Dumping corn stover wastes valuable resources and causes pollutant emissions to the atmosphere and ground water. Methods to utilize crop stover include straw recycling, straw materialization, stockfeed, and straw biorefinery (cellulosic ethanol, cellulosic lactic acid, chemical feedstock, and bio-based materials). Among these methods, straw biorefinery is the most promising and value-added utilization [7]. However,

the fermentable sugars in lignocellulose hydrolysate are not readily fermented during LA production because the associated phenolic substances and aldehydes negatively affect microbial growth and reproduction [7,8].

Continuous fermentation via immobilized cells has been used to solve this problem. The use of immobilized cells can effectively reduce the cell load and provide a more robust cell population to improve the utilization of glucose after the enzymatic digestion of lignocellulose [9]. The cell immobilization technique is based on the principle of encapsulating microorganisms in polymer carriers [10,11]. This technology is usually used to study the colony effects of microbes [12]. Calcium alginate, low actuated gellan gum [13], and other polysaccharides are often used as immobilization materials in culture and fermentation systems [14]. However, the poor mechanical performance and low embedding efficiency of these materials limit their industrial applications [15–17].

The cross-linkable F127-dicarbamate methacrylate (F127-BUM), which is widely used in direct writing or extrusion three-dimensional printing in tissue engineering and biotechnology (Scheme 1), has several advantages as a carrier substrate for immobilized cells [18]. This multifunctional polymer can stimulate reactions, including those that occur under changes in temperature and pressure [19]. This feature enables immobilized cells and other substances in the medium to mix evenly [20]. The modification of polymer chains in a light-induced reaction contributes to their cross-linking ability [21], resulting in a tough and flexible material [22]. To date, F127-BUM has been used in the development of a biosensor [23], and in several biotechnological processes involving microorganisms, e.g., modular bio-manufacturing based on co-culture systems and chemical production by multi-kingdom microbial networks in modular bioreactors [24–26]. Several studies have shown that F127-BUM is a promising material in continuous fermentation systems [27].

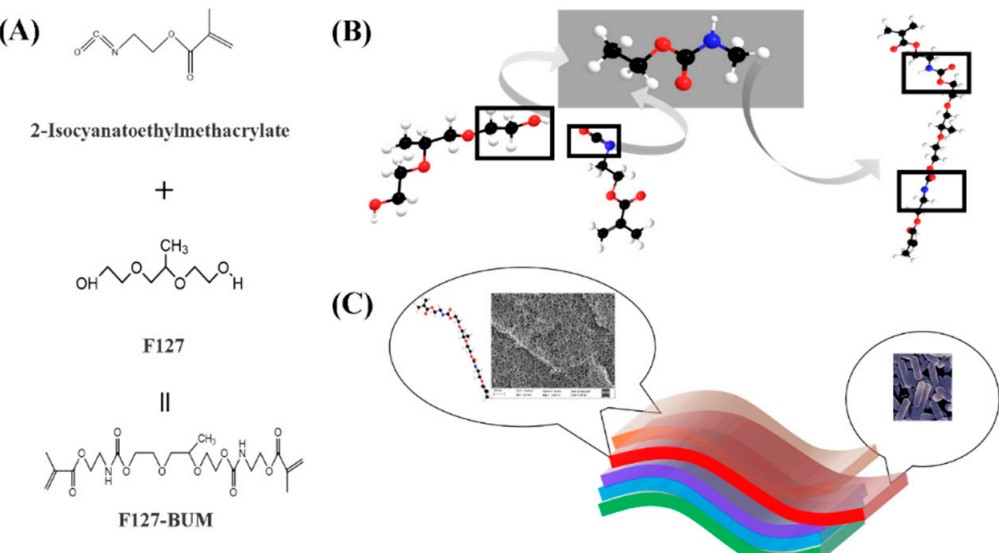

**Scheme 1.** Polymerization of Ethyl methacrylate and Poloxam-F127 (**A**) Chemical synthesis of F127 (**B**) Molecular structure of F127-BUM (**C**) an outer lattice microvesicle which the cells are effectively immobilized in a lat-tice-like structure.

In this study, F127-BUM was used to immobilize cells for the production of D-LA in a continuous fermentation system. Firstly, *Lactobacillus bulgaricus* cells were embedded in F127-BUM. Secondly, the properties of F127-BUM immobilized cells were characterized. Finally, the continuous cell recycling fermentation process using glucose and corn stover hydrolysate as a carbon source for D-LA production was established (Scheme 2).

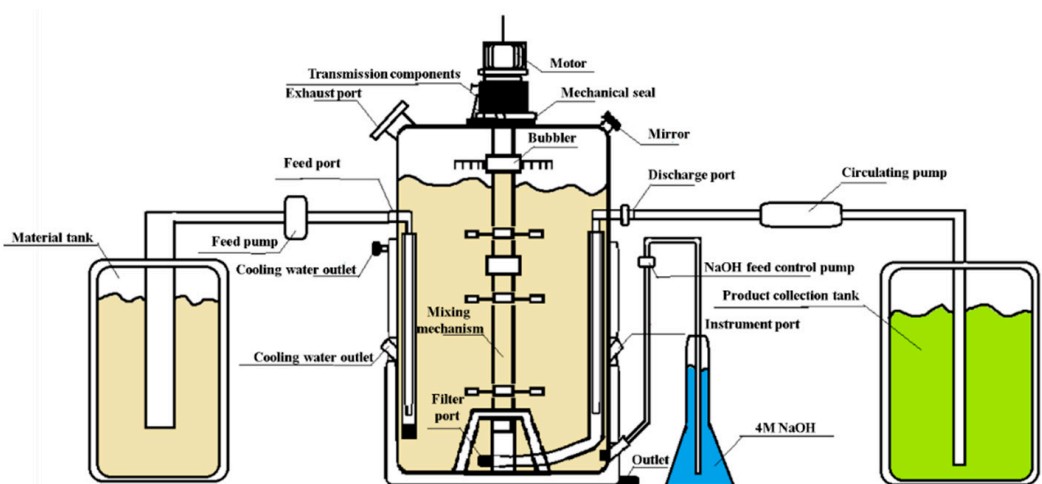

**Scheme 2.** Schematic diagram of continuous Fermentation by immobilized *Lactobacillus bulgaricus* T15 Cells.

## 2. Materials and Methods

### 2.1. Bacterial Strains, Culture Media, and Growth Conditions

*L. bulgaricus* strain T15, herein after referred to as T15, was provided by the Key Laboratory of Straw Comprehensive Utilization and Black Soil Conservation, Ministry of Education, China. The activation medium was MRS broth (Qingdao Haibo Biotechnology Co., Ltd., Qingdao, China). The fermentation medium was MRS broth supplemented with 8% $w/v$ $CaCO_3$. Cells of T15 were cultured without agitation in MRS medium at 37 °C for 24 h until the $OD_{600}$ was between 0.8–1.2

### 2.2. Chemicals and Analytical Instruments

Pluronic F127 was obtained from Sigma (St Louis, MO, USA), and $C_7H_9NO_3$ was obtained from Shanghai Yuan-ye. The glucose detection kit was purchased from the Shanghai Ronghe Biopharmaceutical Industry Co. Ltd. (Shanghai, China). The absorbance of solutions was measured using a Fluostar Omega microplate reader (BMG Labtech, Ortenberg, Germany). Compounds of interest were separated, detected, and quantified by high performance liquid chromatography using a Waters 1525 instrument (Waters, Milford, MA, USA). All reagents were of analytical grade.

### 2.3. Chemical Synthesis of Pluronic F127-BUM

A 30-g portion of F127 (4.8 mmol) was dried at room temperature for 16 h under vacuum (~2 Pa) in a dry glass reactor. Then, anhydrous $CH_2Cl_2$ (250 mL) was added to the reactor under an $N_2$ environment. The temperature was set to 30 °C, and then the reaction catalyst dibutyltin laurate was performed adopting a rate of six drops per second using a glass Pasteur pipette. Next, 2-isocyanatoethyl methacrylate (25 mL) was diluted in anhydrous $CH_2Cl_2$ (50 mL), and this solution was added to the reaction mixture at a rate of approximately one drop per second. The reaction was allowed to proceed while stirring under dry $N_2$ at 30 °C [28]. After 2 days, the reaction was terminated by addition methanol (30 mL). The mixture was then concentrated in a rotary evaporator at 30 °C, before the concentrated solution was added to 1 L ether ($Et_2O$) and the mixture was stirred for 15 min. The F127-BUM was precipitated with $Et_2O$, with stirring, in a large conical flask. The concentrate was poured in slowly, and then, after stirring and precipitation, the mixture was centrifuged at 3000 rpm for 10 min. The transparent supernatant was removed, and more of the precipitate mixture was added on top of the F127-BUM pellet. This process was repeated once. The F127-BUM precipitate was then washed twice. The whole process was repeated. The excess ether was allowed to evaporate while agitating the F127-BUM with a spatula under an $N_2$ atmosphere. The resultant F127-BUM powder was dried fully

overnight at room temperature under vacuum (~2 Pa) and stored in the dark at 4 °C until further use (Scheme 3).

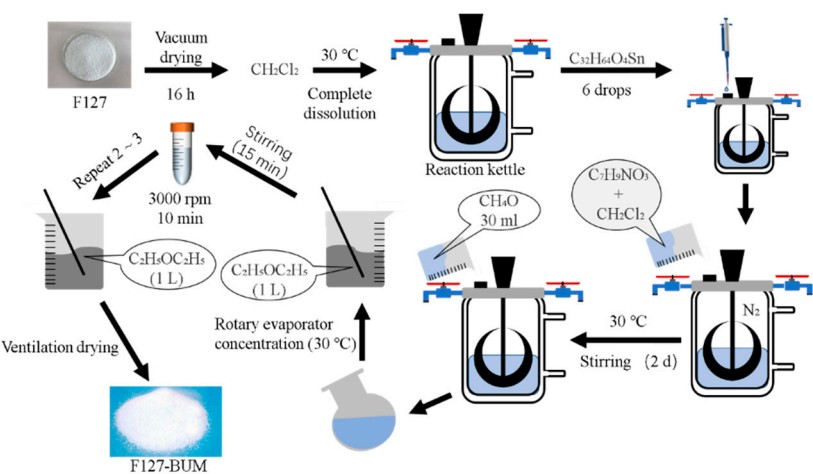

**Scheme 3.** The preparation process of F127-BUM.

### 2.4. F127-BUM Generation and Processing

The dried F127-BUM (3 g) was dissolved in sterile deionized water (7 g) and the resulting mixture was kept overnight at 0 °C to allow complete dissolution of the polymer. The solution was warmed to room temperature (28 °C) to facilitate gel transformation, and then transferred into an ice bath. The photoinitiator 2-hydroxy-2methylpropiophenone (10 μL per 10 g hydrogel solution) was then added to the gel. After thorough mixing, the hydrogel was exposed to UV 365 nm light (at 3.4 mW cm$^{-2}$) to cure for 1 h (Scheme 4) [28].

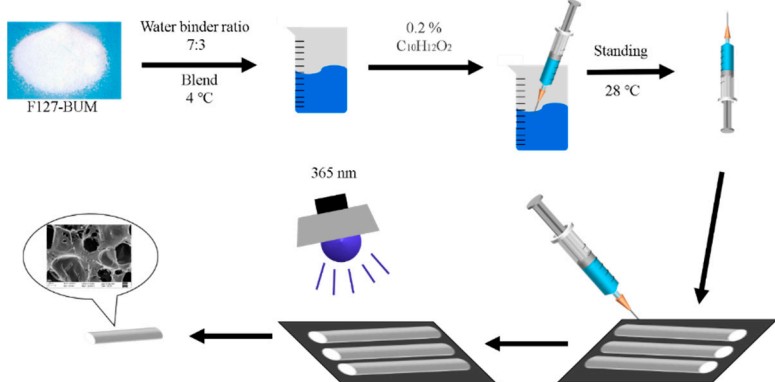

**Scheme 4.** The preparation of the colloidal F127-BUM.

### 2.5. Immobilization of Lactobacillus T15 Cells in F127-BUM Hydrogel

The following steps were conducted at 28 °C. The free radical initiator and activated T15 seed solution were added to F127-BUM at a ratio of 1:10, and then the mixture was transferred to a 25-mL sterile injector to prepare microcapsules. The gel was cured under UV light (365 nm) for 1 h. The prepared microcapsules were activated at pH 6.5 with MRS medium for 24 h. Then, the conditions were further optimized to determine the optimal parameters.

### 2.6. Analyses of Physicochemical Properties of F127-BUM
2.6.1. SEM Analysis of Material Structure

High vacuum field emission scanning electron microscopy was used to characterize F127-BUM microcapsules. The morphology and surface of the capsules before and after immobilization were observed by SEM (EHT 10.00 kV, Signal = inLens) after the samples were treated with gold plating.

2.6.2. Mechanical Properties Analysis of F127-BUM Colloid

A TA-XT 2I texture analyzer (Stable Micro Systems, Godalming, UK) was used for the uniaxial stress compression tests [29]. The fixture parameters were a pressure drop rate of 1 mm/s and compression strain of 80%. The main parameters used for the mechanical strength analysis were strain break, stress break, and Young's modulus. Because the cross-sectional area of the microcapsules changed during the test, corrections were made according to Equations (1) and (2), representing the c-model of p/36R and the compression test mode. The specific test results were translated into Hencky stress (σH) and Hencky strain (εH), respectively.

$$\sigma H = \frac{F_{(t)} \cdot H_{(t)}}{(H_0 \cdot A_0)} \tag{1}$$

$$\varepsilon H = -\ln \left[ \frac{H_{(t)}}{H_0} \right] \tag{2}$$

where $F_{(t)}$ is stress at time $t$ in N; $H_{(t)}$ is the height of the sample at time $t$ in mm; $A_0$ is the initial cross-sectional area of the sample in mm$^2$; and $H_0$ is the initial height of the sample in mm. The fracture stress and fracture strain are the σH-ε stress and strain corresponding to the highest point of the H curve. Young's modulus is σH-ε, the slope of the linear part at the beginning of the H curve.

*2.7. Determination of Optimum Conditions for Immobilized Cell Fermentation*

To determine the optimum fermentation conditions, we tested inoculum amounts of 1, 2, 4, 6, and 8%($v/v$), initial glucose concentrations of 60, 70, 80, 90, and 100 g/L, and temperatures of 37, 39, 41, 43, and 45 °C. An orthogonal experimental design was used to further determine their optimal combination.

*2.8. Analysis of Tolerance to Phenolic Substances and High Concentrations of Glucose*

We analyzed the tolerance of immobilized T15 cells to phenolic substances, with free T15 cells as the control. In these analyses, phenolic substances (5-hydroxymethylfurfural (5-HMF), ferulic acid (FA), vanillin) were added to the medium to final concentrations of 0, 0.5, 1, 1.5, and 2 g/L. In the sugar tolerance experiment, glucose was added to the medium to final concentrations of 60, 70, 80, 90, and 100 g/L.

*2.9. Repeated Fermentation by Immobilized Cells*

The same immobilized cell capsules were used for batch fermentation at the optimum temperature determined in the experiments described in Section 2.7. Every 12 h, the fermentation broth was poured out and replaced with fresh medium. Repeated fermentation was conducted in this way for 100 days. After each fermentation was completed, the hydrogel with immobilized cells was recovered and washed with sterilized deionized water, and subsequently added to fresh fermentation medium for the next fermentation. The yield of hydrogel with immobilized cells in each cycle was monitored to calculate the wear rate.

After 70 d of continuous fermentation, F127 BUM capsules were lyophilized using liquid nitrogen and stored at −80 °C for 30 d. After 30 d, the capsules were removed and inoculated in MRS medium at 37 °C for reactivation, followed by continuous cyclic fermentation for 30 d. At the end of each cycle, 500 μL fermentation broth was collected and the amounts of D-LA and residual glucose were measured. The fermentation process after reactivation was consistent with that during the first 70 d.

*2.10. Enzymatic Hydrolysate of Corn Straw as the Sole Carbon Source for D-LA Production*

The corn stover pretreatment process was referenced our previous study [30]. The optimal condition is 6% NaOH and 12% urea as catalyst, 9% mass fraction of corn stover (pellets 40 mesh) at 80 °C, treated for 20 min, filtered, with and without water washed, and dried at 40 °C for use. The enzymatic hydrolysis process was as follows, take the above pretreated straw, enzyme addition 1.78 mg/mL, temperature 50 °C, 180 rpm for 72 h. After

enzymatic digestion, the enzymatic solution was centrifuged at 3000 r/min for 5 min, the supernatant was collected, and the glucose content was detected.

### 2.11. Determination of D-LA by HPLC

The samples were filtered through 0.2-μm nylon syringe filters (Wheaton Science, Millville, Worcester, MA, USA) before HPLC. Tyrosine and D-LA were detected using a Waters 1525 instrument equipped with an Astec CLC-L column (15 cm × 4.6 mm) (Supelco, Bellefonte, PA, USA) with the detection wavelength set to 245 nm. The constant flow rate was set at 1.2 mL min$^{-1}$ at 25 °C. Tyrosine and acetic acid were detected and quantified using a Waters 1525 instrument equipped with a PRONTOSIL 120-10-C18 H column (250 × 4.6 mm) (Bischoff, Leonburg, Germany) with detection wavelength set to 210 nm. The constant flow rate was set at 0.5 mL min$^{-1}$ at 15 °C.

## 3. Results

### 3.1. Microstructure and Mechanical Strength of F127-BUM Hydrogel

The microstructures of F127-BUM hydrogel with and without T15 cells were observed by SEM (Figure 1). The F127-BUM hydrogels without T15 cells exhibited a grid-like structure with tightly arranged layers (Figure 1A–C). The F127-BUM hydrogels with embedded T15 cells had a more compact grid-like structure. These results suggested that the structure of the hydrogels is extremely suitable as a carrier for immobilized T15 cells.

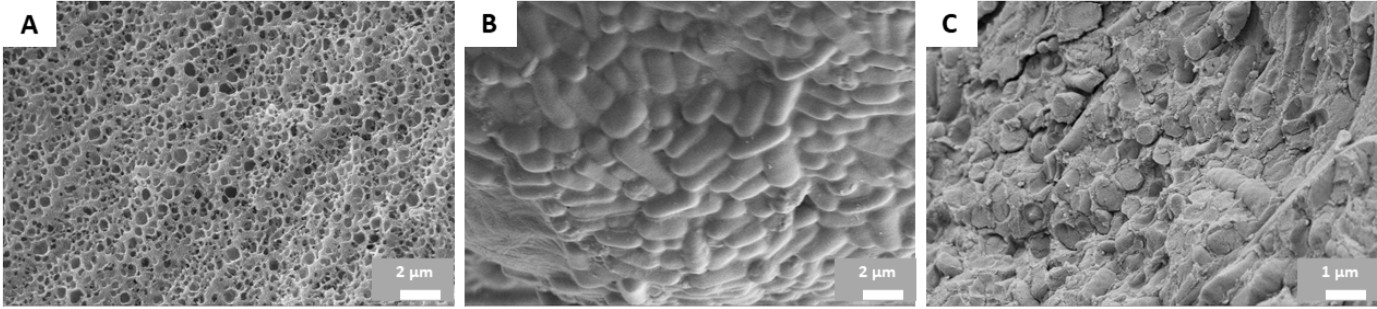

**Figure 1.** SEM of F127-BUM colloids with and without T15. F127-BUM colloids (**A**), Surface microtopography of F127-BUM colloids with T15 immobilizztion (**B**), Sectional view of F127-BUM colloids with T15 immobilization (**C**).

The mechanical strength of F127-BUM hydrogel capsules was determined using a texture analyzer. The Hencky stress was 159.11 kPa and the Hencky strain was 0.646 (Table 1). These values were better than those of most conventional immobilization materials, providing further evidence that F127-BUM hydrogel is suitable for immobilization of T15 cells.

**Table 1.** The microscopic lattice structure and mechanical strength of Calcium alginate, LA-GAGR and F127-BUM-T15.

| | Capsule Diameter (mm) | Aperture Size (μm) | Stressed (N) | ELT (Pa) | Hencky Stress (kPa) | Hencky Strain | WHC (%) |
|---|---|---|---|---|---|---|---|
| Calcium alginate 0.75% (*v/v*) | 2.5 ± 0.28 | 19.625 ± 0.43 | 1654.33 ± 26.41 | 61.77 ± 8.87 | 40.725 ± 0.29 | 0.66 ± 0.33 | 0.324 ± 0.74 |
| LA-GAGR 1.8% (*v/v*) | 2.5 ± 0.75 | 0.977 ± 0.77 | 1922 ± 25.98 | 146.38 ± 27.99 | 91.684 ± 0.11 | 0.67 ± 0.43 | 0.64 ± 0.82 |
| F127-BUM-T15 30% (*v/v*) | 2.5 ± 0.48 | 0.534 ± 0.31 | 5818 ± 16.36 | 258 ± 10.59 | 159.112 ± 0.18 | 0.646 ± 67 | 0.856 ± 0.47 |

### 3.2. Optimal Fermentation Conditions for D-Lactic Acid Production by Immobilized T15 Cells

The D-LA production of immobilized T15 cells remained at a high level up to an initial glucose concentration of 90 g/L. At this concentration, nearly all the free T15 cells were

non-viable (Figure 2A). These results indicated that immobilized T15 cells can tolerate a high glucose concentration. The D-LA yield via T15 cells immobilized in F127-BUM was 57.17 ± 1.39 g/L with an initial glucose concentration of 90 g/L. When the initial glucose concentration exceeded 90 g/L, the D-LA production decreased with increasing glucose concentration (Figure 2A). This may be because a high concentration of glucose negatively affected bacterial growth and metabolism. The size of the inoculum is another important factor affecting D-LA yield. The highest yield of D-LA (56.63 ± 0.61 g/L) was achieved with 6% inoculum, with 70.78% sugar-to-acid conversion ratio. The acid yield decreased with increasing inoculum size, and the sugar-to-acid conversion ratio decreased further when the inoculum size was 8%. This is probably because most glucose was used to produce bacteria so there was insufficient remaining for acid production. Overall, an inoculum of 6% produced the best fermentation results (Figure 2B). The optimum temperature for D-LA production did not change after cell immobilization. As the reaction temperature increased, the yield of D-LA first increased and then decreased, and with the maximum yield at 41 °C (Figure 2C). In conclusion, the optimal fermentation conditions for D-LA production by F127-BUM-immobilized T15 cells were 6% inoculum, 90 g/L initial glucose concentration, at 41 °C in static culture.

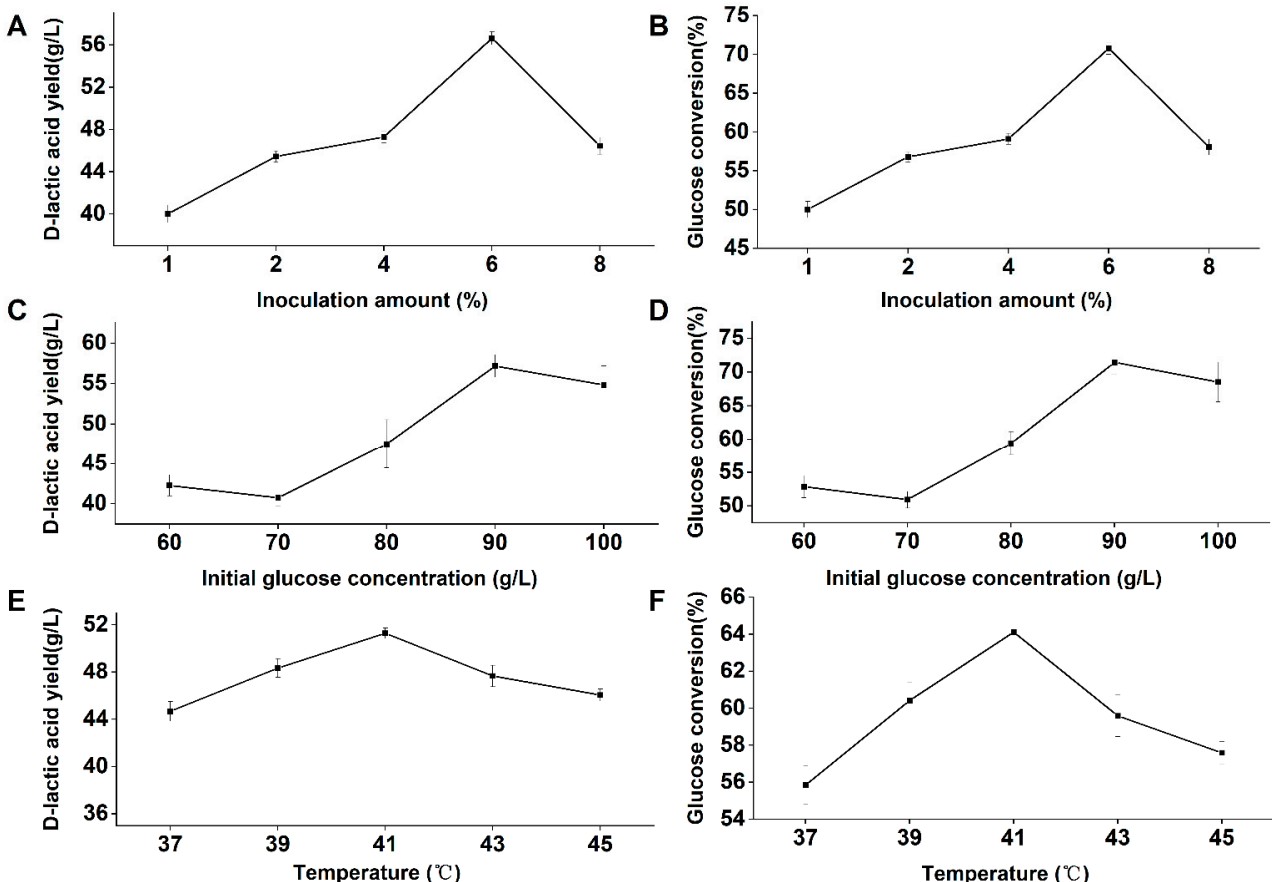

**Figure 2.** Fermentation conditions for D-lactic acid production by immobilized T15 cells (**A**) The effect of inoculation amount on D-LA yield; (**B**) The effect of inoculation amount on glucose conversion; (**C**) The effect of initial glucose concentration on D-LA yield; (**D**) The effect of initial glucose concentration on glucose conversion; (**E**) The effect of temperature on D-LA yield; (**F**) The effect of temperature on glucose conversion).

### 3.3. Tolerance of F127-BUM-T15 to Phenolic Compounds and Glucose

The low tolerance of microorganisms to phenols is one of the main problems to solve in the lignocellulosic biorefinery industry. Our results showed that, after immobilization in F127-BUM, the T15 cells were able to tolerate 5-HMF, FA, vanillin, and high concentrations

of glucose. The yield of D-LA produced by immobilized T15 cells in the presence of a high concentration of 5-HMF was significantly higher than that of free-living T15 cells (Figure 3A,D). The free-living T15 cells produced D-LA from 36 h in the presence of a high concentration of 5-HMF, while the immobilized T15 cells produced D-LA from 12 h. This may be because 5-HMF inhibited the activity of enzymes involved in carbon metabolism on the cell membrane, thus reducing the specific growth rate. Compared with free-living T15 cells, the immobilized T15 cells were less sensitive to aldehyde, and the immobilized cells grew better than the free-living cells. The immobilization of T15 cells had little effect on FA tolerance (Figure 3). However, immobilized T15 cells grew better than did free T15 cells in the presence of a high concentration of FA (Figure 3B,E). Compared with free-living cells, the immobilized T15 cells produced significantly more D-LA in the presence of vanillin (Figure 3C,F). These results showed that F127-BUM is able to protect immobilized T15 cells against toxic substances. The immobilized T15 cells were able to produce D-LA when the initial glucose concentration was high (100 g/L). Thus, the utilization of F127-BUM-immobilized T15 cells for fermentation could improve the equipment utilization rate and reduce production costs (Figure 3G). Compared with fermentation by free-living T15 cells, which requires multiple batches of carbon source supplementation, immobilized T15 cells do not require supplementation, so there is less risk of contamination and less waste of resources.

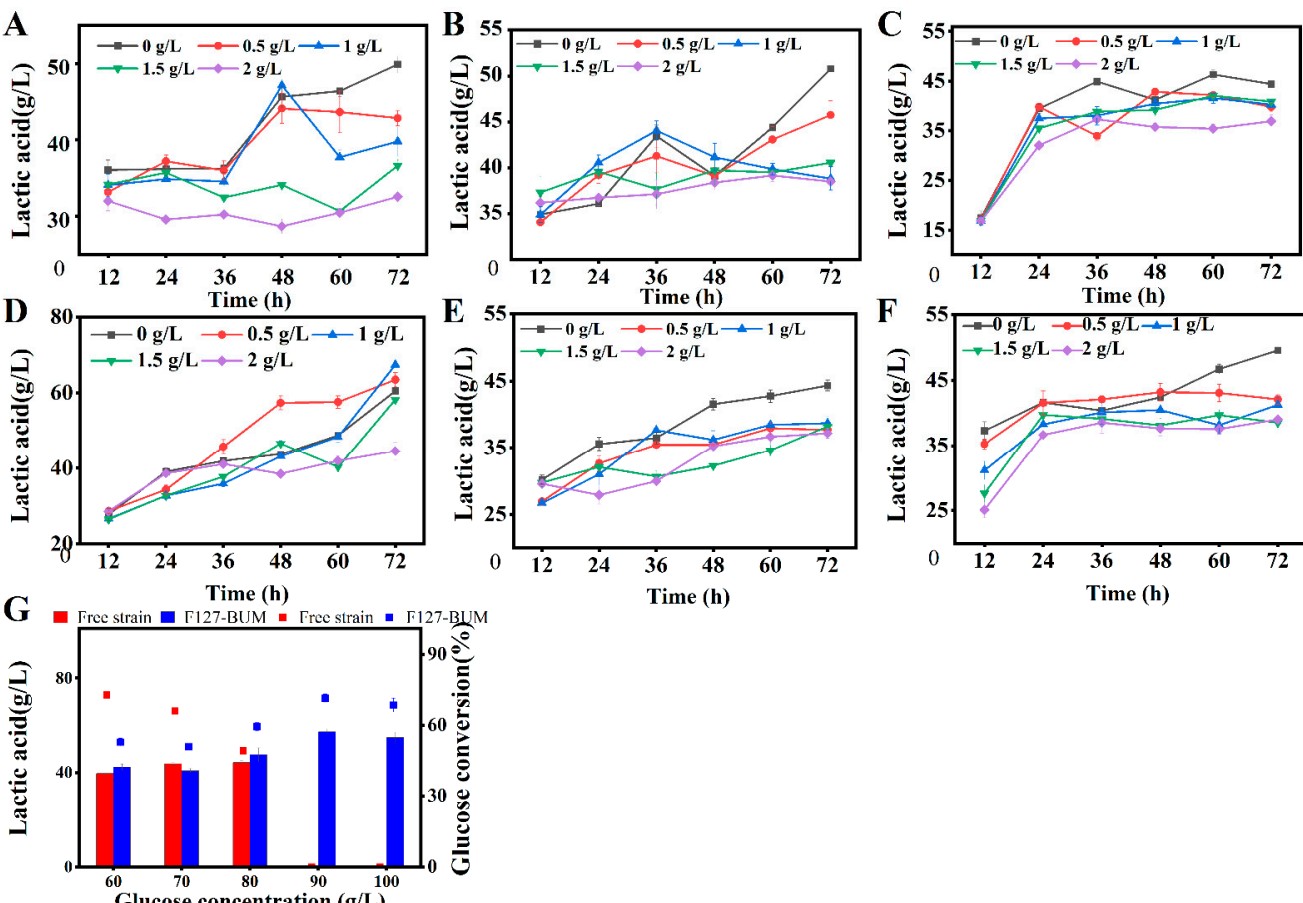

**Figure 3.** Tolerance analysis of 5-HMF by free-living fermentation (**A**) and F127-BUM microencapsulated colloids (**D**), FA by free-living fermentation (**B**) and F127-BUM microencapsulated colloids (**E**), Vanillin by free-living fermentation (**C**) and F127-BUM microencapsulated colloids (**F**), and F127-BUM microencapsulated colloids in a high glucose concentration environment (**G**). Immobilized T15 at different glucose concentrations showed the yield of D-LA and the conversion rate of D-LA and glucose.

### 3.4. Continuous Fermentation via Immobilized T15 for D-LA Production

When glucose was used as the sole carbon source for continuous fermentation, the metabolites included D-LA, acetic acid, and formic acid. After 100 days, the final D-LA yield of immobilized T15 cells was 1982.97 ± 1.81 g at a yield of 2.68 ± 0.48 g/L h, which was higher than that of free cells (0.625 ± 0.28 g/L h), and the glucose-D-lactic acid conversion rate was increased by 28.5% compared to free cells. At 70 days of fermentation, the immobilized T15 cells were collected, lyophilized, and stored at −80 °C. After 30 days, the stored F127-BUM-T15 cells were reactivated by inoculation into MRS liquid medium, and then D-LA continuous fermentation was conducted for another 30 days. The reactivated immobilized T15 cells retained their ability to produce D-LA after storage (Figure 4A). The total D-LA yield was up to 627.45 ± 0.42 g, while the conversion ratio from glucose to D-LA was about 62.2% during the 30 days of fermentation. The longest fermentation time of the F127-BUM-immobilized T15 was about 100 d (Figure 4A). The recoverability of F127-BUM capsules was determined (Figure 4B). The results showed that the wear rate of F127-BUM was less than 10% during the 70-day fermentation period. After lyophilization, the wear rate increased to about 25% but it did not significantly affect D-LA production. To verify the performance of immobilized T15 cells, an enzymatic hydrolysate of corn stover was used as the sole carbon source for D-LA production by continuous fermentation. Two types of enzymatic hydrolysates from with and without water washed pretreated corn stover. The D-lactic acid production from the washed corn stover enzymatic hydrolysate as a carbon source was 43.39 ± 0.88 g/L higher than that of the free T15 cells (34.95 ± 1.68 g/L). The results suggest that the production of D-LA using immobilized T15 cells by continuous cell recycle fermentation has a promising application in the biorefinery industry.

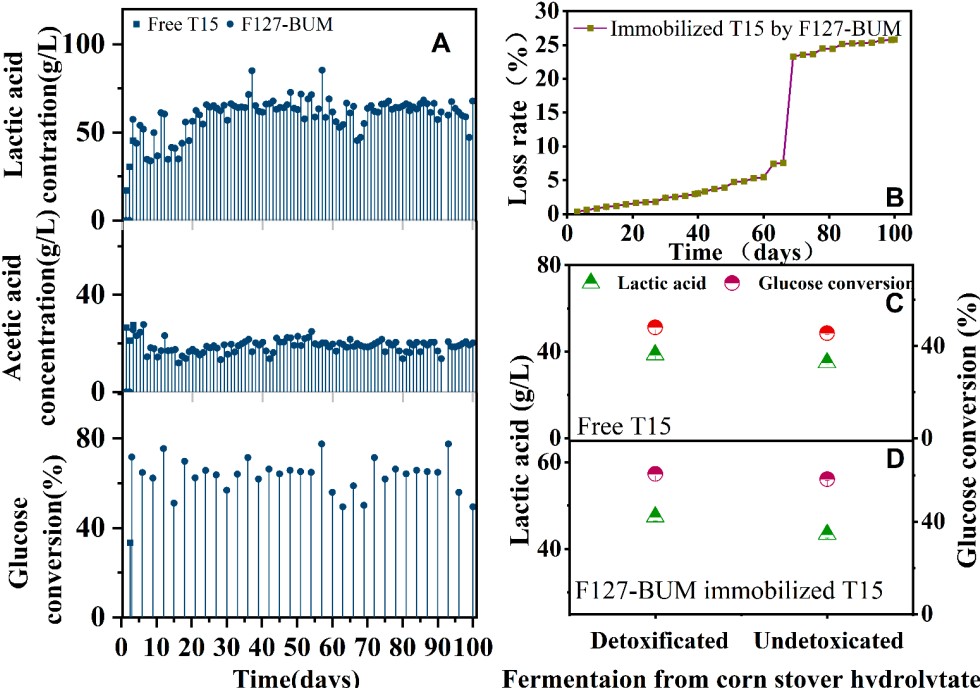

**Figure 4.** (**A**) Yield analysis of D-LA and its by-product acetic acid obtained from the simultaneous extreme fermentation of free-living bacteria, F127-BUM microencapsulated colloids. Activation of the product after 30 d of liquid nitrogen cryopreservation after 70 d and cyclic fermentation for 30 d again to confirm the yield analysis of the activated product (**B**) Comparison of the wear rate of F127-BUM microencapsulated colloids during the extreme fermentation. Corn stover hydrolysate was selected as the sole carbon source for D-LA production. LA production by immobilized LA microencapsulated colloids (**C**): non-hydrolysis detoxification treatment (**D**): hydrolysis detoxification treatment.

## 4. Conclusions

A self-healing hydrogel, F127-BUM, was prepared experimentally and used to immobilize *Lactobacillus bulgaricus* T15 cells for D-LA production by continuous cell recycle fermentation. The optimum fermentation conditions were determined by means of single-factor and orthogonal analysis as an initial glucose concentration of 90 g/L, an inoculum of 6%, and resting culture at 41 °C. Tolerance analysis was also performed for phenols and aldehydes (5-HMF, FA and vanillin) produced by a corn stover pretreatment process. The results showed that the use of F127-BUM immobilized cells improved the tolerance of cells to high concentrations of phenolics and glucose. The total D-LA production via the immobilized T15 was 1982.97 ± 1.81 g, with a yield of 2.68 ± 0.48 g/L h, which was higher than that of free cells (0.625 ± 0.28 g/L h). The subsequent results of single batch fermentation using corn stover hydrolysate as carbon source showed the same conclusion. Continuous cell recycle fermentation via F127-BUM immobilized T15 has excellent potential for application in the biorefinery industry.

**Author Contributions:** G.W. and Y.G. designed this study and wrote the original draft and contributed to the final manuscript. H.C., J.L. and S.Z. reviewed the original draft. Y.L. and M.G. undertook the statistical analyses, prepared the tables and figures. G.C. reviewed the manuscript. All authors have read and agreed to the published version of the manuscript.

**Funding:** The study is funded by Natural Science Foundation of Department of Science and Technology of Jilin Province (20180101268JC) and the Key R&D Plan of Jilin Province (20200402097NC).

**Institutional Review Board Statement:** Not applicable.

**Informed Consent Statement:** Not applicable.

**Data Availability Statement:** All research data were presented in this contribution.

**Acknowledgments:** We thank for the supporting of the Key Laboratory of Straw comprehensive utilization and black soil conservation, Education Ministry of China.

**Conflicts of Interest:** The authors declare no conflict of interest.

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
