# Peer review of "Continuous Fermentation by Lactobacillus bulgaricus T15 Cells Immobilized in Cross-Linked F127 Hydrogels to Produce ᴅ-Lactic Acid"

_fermentation, doi:10.3390/fermentation8080360_

Round 1

Reviewer 1 Report

The present study proposed the idea of using F127-dicarbamate methacrylate to immobilize the cells for D-lactic acid production from corn stover. It is a good study. However, there are some important modifications required in the manuscript before it is accepted for publication.

There is tremendous lignocellulosic biomass available, the author should justify the reason for the selection of corn stover for the present study.

The discussion part is very poor. The results need to be discussed in more detail.

There are many protocols used in the present study without any reference citation.

 Please mention the D-LA yield using free-living cells for better understanding, L23-24, L257

L68, the author should provide the information of media/substrate used for continuous fermentation study using free and immobilized cells

L143, mention the unit of percentage inoculum (1%, 2%, 4%, 6%, 8%).

There is no information on pretreatment and enzymatic hydrolysis concentration in section 2.10

The data mentioned in the manuscript should have a standard deviation.

Author Response

Deer reviewer,

Thank you very much for your comments and suggestions, which are extremely important for our study.

A point-by-point response was upload as a word file. Please see the attachment.

Reviewer 2 Report

Kindly find the following comments and include this in the text to improve the quality of the manuscript

I will recommend that the authors conduct additional experiments in which real-time pretreated lignocellulosic sugars are used as a substrate for immobilised Lactobacillus T15 cells to validate the efficiency of the fermentation using this technique. Only sugar, ferulic acid, and vanillin-containing medium will not provide a clear picture.

The benefits of using non-food lignocellulose materials as a substrate for LA production are underappreciated. The provided justification is not critical because converting food waste to lactic acid is easier and more cost effective than converting lignocellulosic waste. Simultaneously, you will be unable to convert to lactic acid until you have access to sugar. Please provide a detailed explanation.

Vanillin is an antimicrobial agent, hence justifies its use as carbons source in lactic acid production. Mention the vanillin toxic limit for L. bulgaricans at the same time. Please explain why the author used ferulic acid and vanillin to test resistance to phenolic substances when lignocellulosic materials can produce a variety of toxic compounds as well as a wide range of sugars during pretreatment.

Please specify how long the culture was stored at room temperature for subsequent continuous fermentation in materials and methods, section 2.1, last line.

In section 2.5 during Immobilization and before please mention the optimum pH used for the growth of Lactobacillus T15 cells.

Please describe the detailed methodology of SEM analysis in section 2.6.1.

Section 2.7 explains how to optimise fermentation conditions by adjusting all parameters simultaneously (sugar concentration, inoculum concentration, and temperature). Use a design expert-RSM system to optimise the best fermentation conditions.

Author Response

(The authors gave the same response as above.)

Round 2

Reviewer 1 Report

The author has addressed all the raised queries very carefully and modified the manuscript accordingly. The manuscript is accepted for publication in its present form. Thank you!

Good Luck!

Author Response

Dear reviewer,

    Thank you very much for your hard work and detailed comments and suggestions. Your suggestions are not only of great help to this manuscript, but also plays a key role in guiding my later research. Thank you again. 

Gang Wang

JiLin Agricultural University